# Technology availability, sector policies and behavioral change are complementary strategies for achieving net-zero emissions

Jessica Strefler [1] ✉, Leon Merfort [1,2], Nico Bauer [1], Miodrag Stevanović [1], Dennis Tänzler[3], Florian Humpenöder [1], David Klein[1], Gunnar Luderer [1,2], Michaja Pehl[1], Robert C. Pietzcker [1], Alexander Popp [1,4], Renato Rodrigues [1], Marianna Rottoli[1] & Elmar Kriegler [1,5]

In this study, we analyze the effects of technology availability, political coordination, and behavioral change on transformation pathways toward net-zero greenhouse gas emissions in the European Union by 2050. We implemented an iterative stakeholder dialogue to co-design the scenarios that were calculated using a global multi-regional energy-economy-land-climate model. We find that in scenarios without behavioral change and with restriction of technologies, the target of greenhouse gas neutrality in the European Union cannot be reached. Already a target of 200 Mt $CO_2$eq/yr requires $CO_2$ prices above 100 €/t$CO_2$ in 2030 across all sectors in all scenarios. The required $CO_2$ price can increase to up to 450 €/t$CO_2$ by 2030 if technologies are constrained, if no complementary regulatory measures are implemented, and if changes in consumer behavior towards a more sustainable lifestyle do not materialize.

The Sixth Assessment Report of the Intergovernmental Panel on Climate Change (IPCC)[1] has shown that the majority of 1.5 °C scenarios achieve net-zero $CO_2$ emissions globally by mid-century. To adhere to this target, the European Union (EU) adopted the European Green Deal[2,3] and the EU Long-term low greenhouse gas emission development strategy of the European Union and its member states[4] setting the more ambitious goal of net-zero greenhouse gas (GHG) emissions by 2050[5,6]. While the target is accepted and supported among member states, the implementation is still widely debated. For instance, policy measures to achieve especially emission reductions within the Effort Sharing Regulation (ESR) can include bans, taxes, subsidies, or standards, and may vary between member states. The availability of some technologies like carbon capture and storage (CCS), wind power, nuclear power, or bioenergy depends not only on technological development but also on social acceptance and political feasibility. While the impacts of technology options on climate change mitigation strategies have been extensively studied[7–10], more comprehensive studies also including an interplay with different policies and behavioral

changes across all sectors are much less common. In this study, we aim to close this gap by bringing together different technology options, policy measures, and behavioral changes in a coherent scenario set.

Improved and iterative stakeholder dialogue is crucial to improve the scenarios' relevance for policy making and to obtain scenarios that are not only technically, but also politically and socially feasible[11,12]. In recent years, multi-stakeholder engagement including business, government, civil society, and science has become a key ingredient for tackling the challenges of climate change mitigation[13,14]. For this study, we implemented such an iterative stakeholder dialogue process with representatives from the groups mentioned above to first identify the most relevant transformation measures in the buildings, industry, transport, and land sectors. We then grouped the measures across all sectors into (i) technology and innovation, (ii) political coordination, and (iii) behavioral change, which represent three key dimensions of the debate that are emphasized to a different degree by different stakeholders. From these three dimensions of transformation, we derived a set of scenarios that was again vetted with the stakeholders

[1]Potsdam Institute for Climate Impact Research (PIK), Member of the Leibniz Association, Potsdam, Germany. [2]Global Energy Systems, Technische Universität Berlin, Berlin, Germany. [3]adelphi consult GmbH, Alt-Moabit 91, Berlin, Germany. [4]Faculty of Organic Agricultural Sciences, University of Kassel, Kassel, Germany. [5]Faculty of Economics and Social Science, University of Potsdam, Potsdam, Germany. ✉e-mail: strefler@pik-potsdam.de

to capture the most relevant narratives. A more detailed description of the stakeholder engagement is provided in Supplementary Note 1.

Each dimension has two possible realizations, which stand for different possible technological developments, actions in politics, or developments in society. In the "technology and innovation" dimension, we contrast a "focus on GHG mitigation", where all mitigation technology options are available, with a "focus on social acceptance", where technologies with a lack of (perceived) support in the population, such as CCS, nuclear, but also wind power, are restricted. In the "political coordination" dimension, we contrast a "market-oriented" approach, with a cross-sector $CO_2$-eq price as the central measure, with a "sector-oriented" approach, in which the $CO_2$-eq price is complemented by targeted sector policies. These sector policies can be modeled explicitly, e.g. a ban on internal combustion engines for light-duty vehicles or oil and gas heating, or implicitly, e.g. by assuming a lower hydrogen or electricity price as a proxy for subsidies. In the "behavioral change" dimension, it is either assumed that consumer behavior is only based on price signals ("price-oriented") or that there is a broad shift towards low-carbon and more sustainable consumption of goods and services, including modal shifts or changing dietary choices ("value-oriented"). More detail including a list of all transformation measures in the three dimensions is provided in Supplementary Note 2.

In close cooperation with the stakeholders, we derived five scenario narratives from the potential combinations of the different realizations, which represent different approaches to the transformation challenge (Table 1). The "policy steering approach" (S1) assumes that only the subset of technologies that is perceived to be socially and ecologically sustainable is deployed at a large scale. Targeted sector policies complement a carbon pricing scheme, and while the public supports stringent climate policies, they do not change their behavior beyond price signals. The "behavioral approach" (S2) follows a similar narrative but adds behavioral changes toward sustainability. The "technology-oriented approach" (S3) assumes supporting sector policies as well as no behavioral change but allows for the use of the full set of mitigation options. The "acceptance-oriented approach" (S4) and the "market economy approach" (S5) both achieve their climate targets only using carbon pricing mechanisms, without additional sector policies, and without behavioral changes. In terms of technologies, the latter uses the full set of mitigation options while the "acceptance-oriented approach" relies only on technologies perceived to be socially and ecologically sustainable. The narratives are translated to scenarios that aim to be both globally consistent with the 1.5 °C target and meet the EU climate target of net-zero GHG emissions in 2050. The state-of-the-art multi-regional energy-economy-land model with global coverage REMIND-MAgPIE[13–15] is extended and adapted to represent these transformations and derive deep decarbonization pathways for the EU consistent with the net-zero strategy, within a world aiming for a 1.5 °C limit of global mean temperature increase.

In this work, we provide insights into the interplay between policies, technology availability, and behavioral change, and their impact on land and energy systems and on selected economic, social, and environmental indicators informing the feasibility of the net zero transition. We find that in scenarios without behavioral change and with restriction of technologies, the target of GHG greenhouse gas neutrality in the

European Union cannot be reached. Already a target of 200 Mt $CO_2$eq/yr requires $CO_2$ prices above 100 €2020/$tCO_2$ in 2030 across all sectors in all scenarios. The required $CO_2$ price can increase to up to 450 €2020/$tCO_2$ by 2030 if technologies are constrained, if no complementary regulatory measures are implemented, and if changes in consumer behavior towards a more sustainable lifestyle do not materialize.

## Results

### Strong near-term ambition and sufficient carbon prices are necessary

We find that in all five scenarios, GHG emissions in the EU are strongly and immediately reduced (Fig. 1a). However, we find that in scenarios without behavioral change ("Price oriented") but with restriction of technologies like CCS ("Focus social acceptance"), the target of GHG neutrality in the EU could not be reached (S1 and S4). To still be able to compare all scenarios, the climate target was relaxed to 200 Mt $CO_2$eq/yr residual emissions in 2050, which would have to be offset by additional carbon dioxide removal (CDR) options not represented in the model. For comparison, the results of scenarios S2, S3, and S5 achieving the full GHG neutrality target are discussed in Supplementary Note 3.

While the emission trajectories are similar in all scenarios, the underlying transformation pathways differ. The necessary carbon prices in 2030 to achieve the climate target range from 125 €2020/$tCO_2$ (S2) to more than 450 €2020/$tCO_2$ (S4) (Fig. 1b). We find that scenarios with supporting sector policies (S1-S3, red colors) show consistently lower carbon prices than the "market oriented" scenarios (S4-S5, blue colors). Also moving from limited to full technology availability or from price-oriented to value-oriented behavior can both reduce the carbon price to a similar extent. The highest carbon price of 456 €2020/$tCO_2$ in 2030 in the acceptance-oriented scenario S4 can be reduced by about 60% by either full inclusion of all technologies (194 €2020/$tCO_2$ in S5), or by complementing the carbon price with the targeted sector policies (170 €2020/$tCO_2$ in S1). Augmenting the policy steering approach S1 either with full technology availability (S1 to S3) or a value-oriented behavioral change (S1 to S2) both lead to a further significant reduction of carbon prices by about 25-35% from 170 €2020/$tCO_2$ to 125 €2020/$tCO_2$ and 109 €2020/$tCO_2$, respectively. Depending on the scenario, this would mean a moderate to large increase of emission prices in the emissions trading system (ETS) from above 90 €2020/$tCO_2$ in 2020, and a much larger increase of carbon prices in the buildings and transport sector in the new emissions trading system ETS2, which are currently discussed to be capped at 45 €2020/$tCO_2$. Carbon prices in a similar range have been found in previous studies[12,15,16].

### Large-scale deployment of technologies can significantly reduce carbon prices

In the "technology and innovation" dimension, technologies with a (perceived) lack of public support are excluded or restricted, such as wind power, nuclear power, bioenergy, or CCS (scenarios S1, S2, S4). Limiting CCS implies a limitation of carbon dioxide removal (CDR), which is mainly supplied via bioenergy with CCS (BECCS). Direct air capture with CCS (DACCS) is available, but not used in the model in the

**Table 1 | Scenarios are built from different combinations of the three dimensions of emission reduction measures**

| | Scenario Name | Technology & Innovation | Political Coordination | Behavioral Change |
|---|---|---|---|---|
| S1 | Policy steering approach | Focus social acceptance | Sector oriented | Price oriented |
| S2 | Behavioral approach | Focus social acceptance | Sector oriented | Value oriented |
| S3 | Technology-oriented approach | Focus GHG mitigation | Sector oriented | Price oriented |
| S4 | Acceptance-oriented approach | Focus social acceptance | Market oriented | Price oriented |
| S5 | Market economy approach | Focus GHG mitigation | Market oriented | Price oriented |

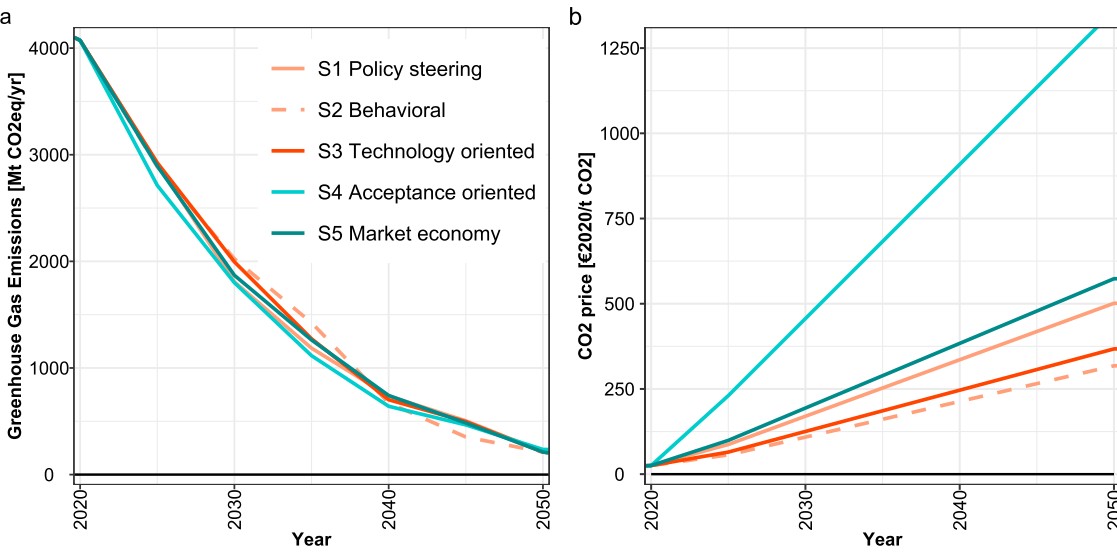

**Fig. 1 | Key characteristics of transformation scenarios. a** Annual greenhouse gas emissions and **b** carbon prices in the European Union for all five scenarios. Red colors represent the sector-oriented scenarios S1-S3, blue colors market-oriented scenarios S4-S5, with full (S3, S5) or restricted (S1, S2, S4) technology availability. See Table 1 for the description of scenarios S1-S5.

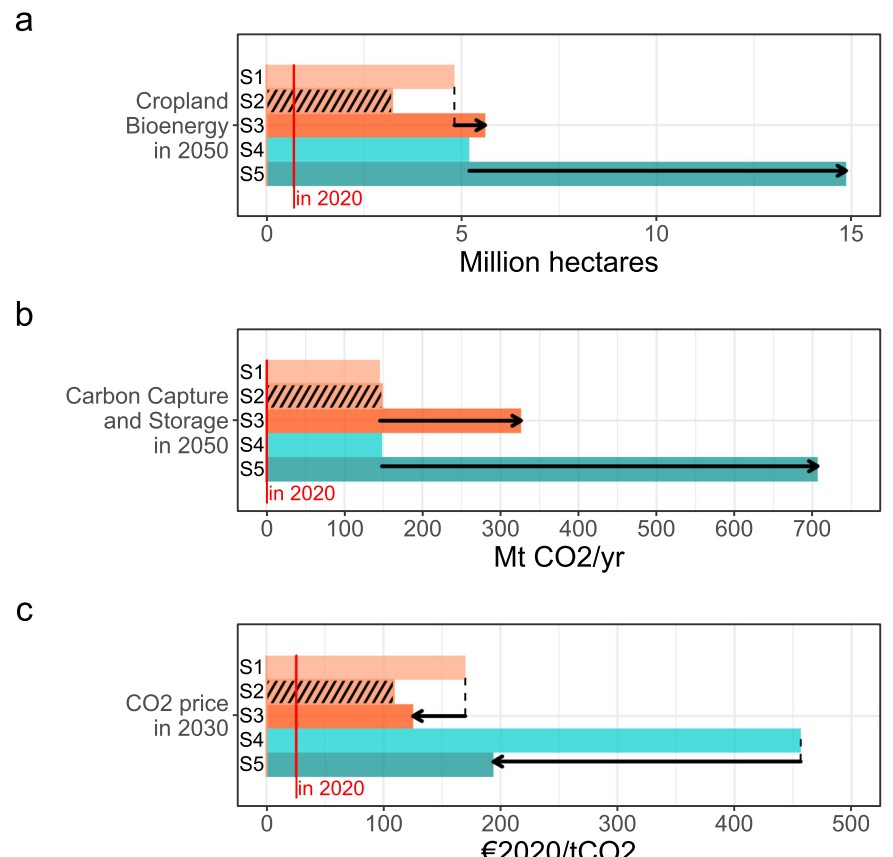

**Fig. 2 | Key trade-off for the technology and innovation dimension in the European Union. a** Cropland for bioenergy and **b** carbon capture and storage (values shown for 2050) increase, while **c** carbon price (values shown for 2030) decreases. We also indicate the respective values in 2020. Black arrows indicate the trade-off between limited vs. full technology availability for sector-oriented (S1 vs. S3) and market-oriented (S4 vs. S5) scenarios. See Table 1 for the description of scenarios S1-S5.

EU28 before 2050 due to higher costs. Lower CDR availability leaves less leeway for residual emissions requiring more costly mitigation measures and therefore increasing the carbon price. This carbon price increase is higher in "market oriented" scenarios (135% from 194 €2020/tCO$_2$ in S5 to 456 €2020/tCO$_2$ in S4) than in "sector oriented" scenarios (36% from 125 €2020/tCO$_2$ in S3 to 170 €2020/tCO$_2$ in S1). The additional sector policies already reduce the residual emissions and therefore reduce the reliance on CDR.

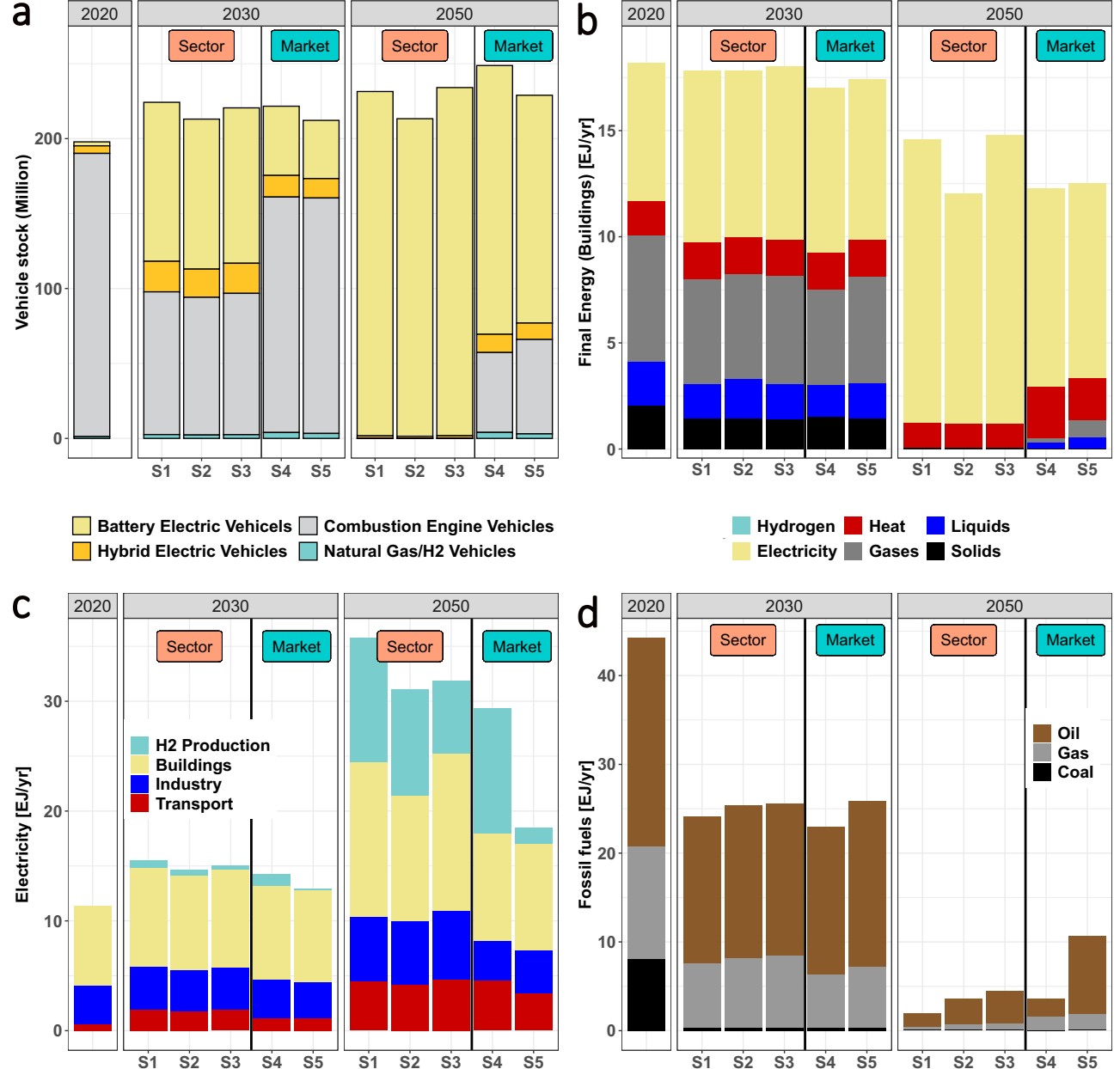

**Fig. 3 | Impacts of sector policies on the energy system in the European Union in 2020, 2030, and 2050. a** Licensed light duty vehicles by mode for all five scenarios. In the sector-oriented scenarios S1-S3 no new internal combustion engines (grey) are permitted after 2030, leading to a phase-out until 2050. **b** Final energy demand in the buildings sector by source, showing higher electrification (yellow) and phase-out of liquids (blue) and gases (grey) by 2050 in the sector-oriented scenarios S1-S3. **c** Electricity demand by sector and **d** fossil fuel use by source. Energy is given in exajoules per year. See Table 1 for a description of scenarios S1-S5.

The fundamental trade-off in the "technology and innovation" dimension is between the acceptance of technologies such as CCS and larger use of agricultural land for bioenergy supply on the one hand, and a more than doubled $CO_2$ price on the other hand (Fig. 2). CCS is used not only for mitigation of industry process emissions, but also for CDR via BECCS. Higher CDR potentials allow for some residual fossil fuel emissions, especially in the industry and transport sectors (see Supplementary Fig. 3a). This reduces the necessary $CO_2$ price, but requires a higher CCS deployment and larger land areas allocated to bioenergy production (Fig. 2). Bioenergy imports are not allowed in the scenarios to avoid land-use change emissions (see Methods section). This trade-off can be mitigated by targeted sector policies, as they reduce residual emissions and therefore the need for CDR requiring CCS and bioenergy.

## Targeted sector policies can reduce the $CO_2$ price

With regard to "political coordination" we analyze the effects of additional sector policies leading to faster uptake of electro-mobility (Fig. 3a) and a phase-out of fossil fuels in the transport and buildings sectors (Fig. 3b). Though these policies lead to significant emission reductions (scenarios S1-S3), they are not sufficient to achieve the climate targets and still rely on being complemented by high carbon prices to cover all sectors.

These targeted sector policies lead to more direct electrification. Phasing out internal combustion engines for light-duty vehicles leads to almost complete electrification of this transportation market segment by 2050 (Fig. 3a). In our scenarios, this cannot be achieved with carbon prices alone, even at more than 450 €2020/t$CO_2$ in 2030 as in scenario S4. Instead, at these high carbon prices the share of energy

that is not directly electrified is decarbonized via indirect electrification using hydrogen or synthetic fuels. This increases hydrogen demand beyond already high demands from the industry sector, which raises questions regarding scalability and adds to the electricity demand.

In line with Luderer et al.[10], enhanced electrification (Fig. 3c) reduces the demand for fossil fuels (scenarios S1-S4 in Fig. 3d) and therefore reduces residual emissions as well as CDR demand (scenarios S1–S4 in Supplementary Fig. 3). At the same time, energy security is increased due to lower dependence on oil and gas (Fig. 3d) which can reduce import dependencies. Coal use is phased out almost completely by 2030, and oil use is reduced by 85-94% by 2050 compared to only 63% in scenario S5 (see also Supplementary Fig. 7). However, a higher share of electrification requires more electricity and therefore faster expansion of renewable energy (scenarios S1-S4 in Fig. 3c). In these scenarios, wind energy annual new capacities would need to be about 4 times higher in 2030 than today (~70-85 GW/yr, Supplementary Fig. 8a), and 10-11 times higher for solar energy (>100 GW/yr, Supplementary Fig. 9a).

The reduction in CDR demand alleviates the trade-off between cropland for bioenergy and CCS, and high carbon prices seen for the technology dimension (Fig. 2). As many of the supporting sector policies involve regulatory policies, the necessary carbon price to achieve the remaining emission reductions is decreased, even though overall economic costs may be higher. The fundamental trade-off between the two approaches is therefore primarily one of social acceptance and political feasibility. A very high $CO_2$ price is accompanied by distributional effects that can jeopardize social acceptance and political feasibility[17,18]. These regressive distributional effects could be mitigated or even made progressive, e.g. by appropriate redistribution of revenues from the $CO_2$ pricing[18,19]. In the present scenarios, targeted sector policies can significantly reduce the $CO_2$ price needed to achieve the climate targets. However, they require stringent regulatory measures such as bans, which must be socially accepted and politically implemented and are associated with distributional issues and legal risks. Yet some hard-to-abate emissions show little response to even very high $CO_2$ prices,

which suggests that a $CO_2$ price in combination with sector policies could be a more politically feasible option, even though it may not maximize economic efficiency.

## Lifestyle changes have multiple benefits

In the "behavioral change" dimension we compare a scenario where consumer choices are purely price-based (S1) with a scenario including value-based consumer choices towards more sustainable options (S2) (see Supplementary Table 1 for specific assumptions). These value-based choices include dietary changes away from animal products towards more vegetables and nuts following the suggestions of the EAT-Lancet commission[20], lower energy consumption in households, and a switch from private cars towards public transport or bikes at scales similar to those discussed in the earlier literature[21–24].

Comprehensive lifestyle changes result in significantly lower non-$CO_2$ emissions from agricultural production (Supplementary Fig. 3), leading to a lower demand for $CO_2$ removal from the atmosphere. The lower non-$CO_2$ emissions reduce the $CO_2$ price required to achieve the climate target by one-third (Fig. 1b). Due to the reduced demand for animal products, less pasture is needed so that they can be rededicated to cropland (Fig. 4) leading to lower food prices. Other additional benefits include positive environmental and health effects associated with agricultural production or diets that are less focused on animal products, as well as lower consumer prices mainly due to the lower $CO_2$ price (Fig. 5). However, the low pressure on food systems only arises in combination with the assumption that energy produced from biomass will be significantly curtailed due to lack of social acceptance. If technologies to remove $CO_2$ from the atmosphere were fully supported and there was high demand for bioenergy crops, such low-intensity agriculture would not be possible, and new trade-offs due to land-use competition could arise.

## Discussion

While all scenarios overachieve the EU climate target for 2030, they differ substantially in their challenges. Figure 5 summarizes the trade-offs between the three dimensions of transformation "technology & innovation", "political coordination", and "behavioral change".

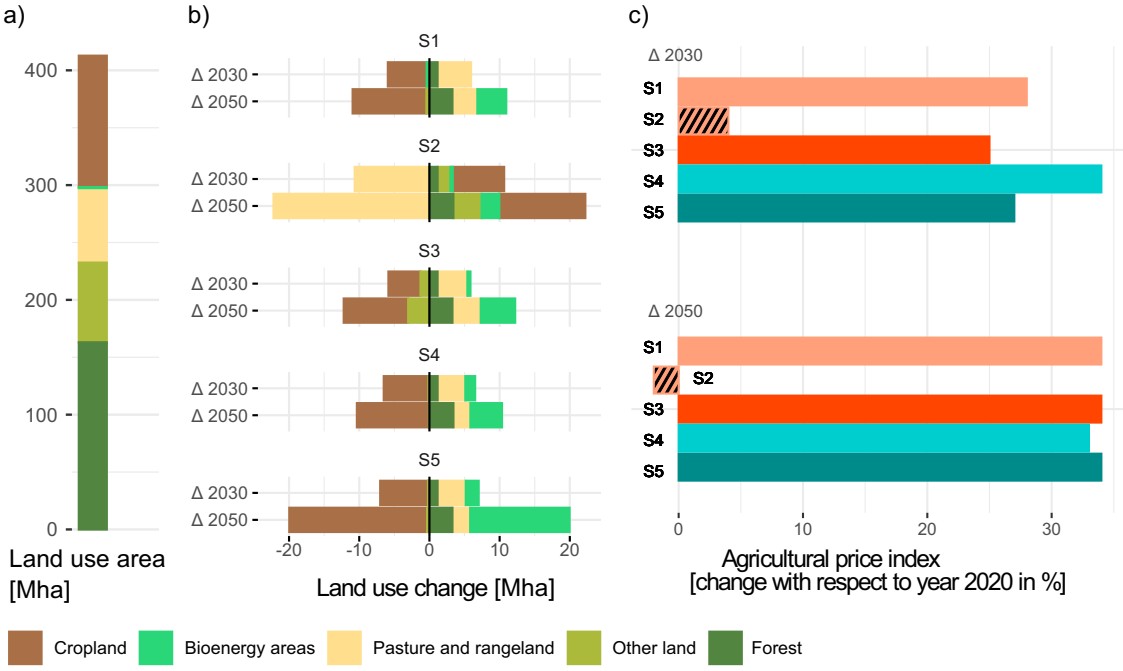

**Fig. 4 | Impacts of behavioral changes on the land-use sector in the European Union. a** Land area in different land use pools in 2020 (cropland (brown) 114.5 million hectares (Mha), bioenergy areas (bright green) 0.7 Mha, pasture and

rangeland (yellow) 65 Mha, other land (olive green) 69.4 Mha, forest (dark green) 163 Mha). **b** Change of land use in 2030 and 2050 and **c** Agricultural price index in 2030 and 2050 compared to 2020. See Table 1 for a description of scenarios S1-S5.

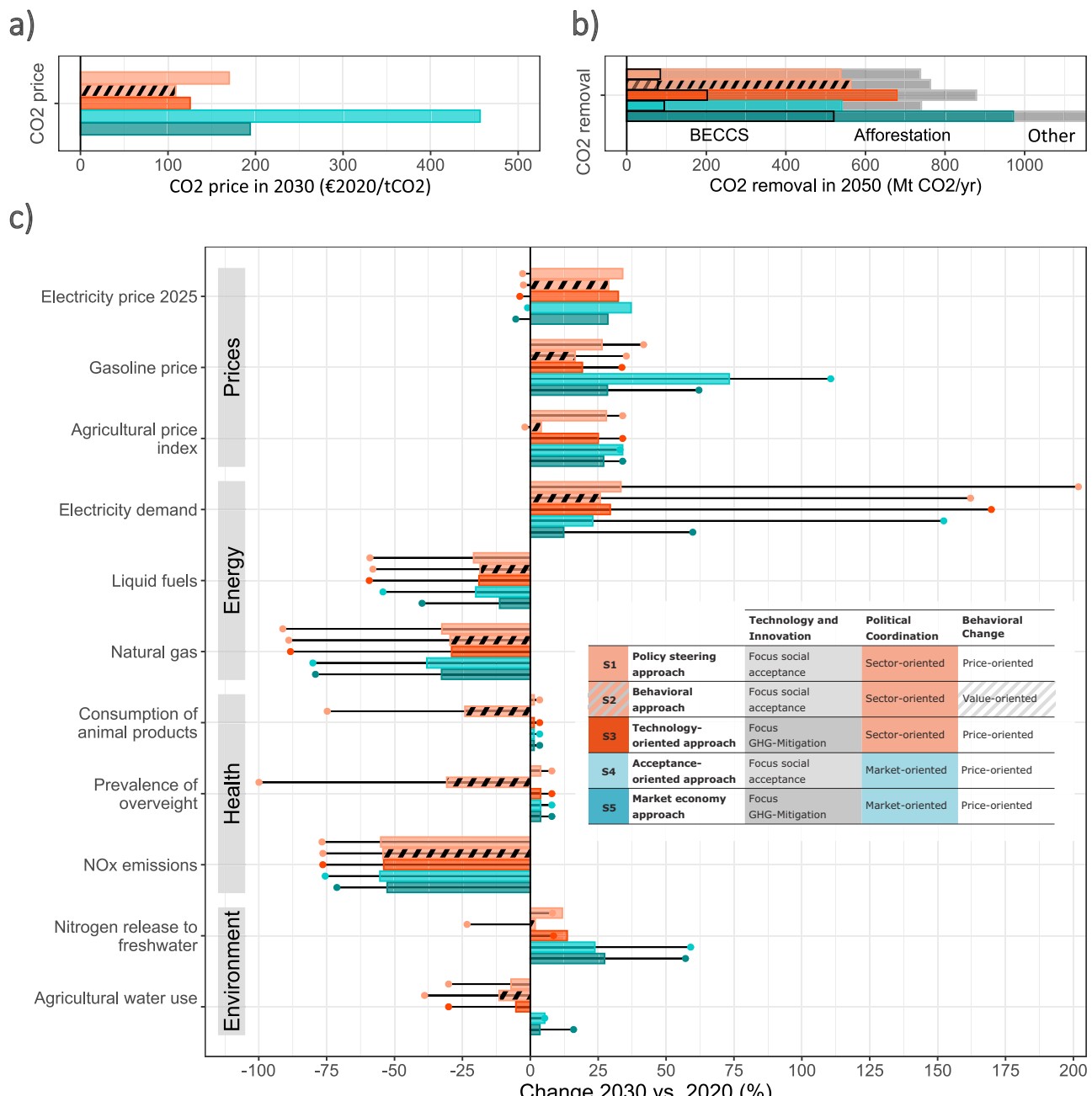

**Fig. 5 | Transformation indicators for all five scenarios.** The colored bars show **a** the $CO_2$ price in 2030, **b** carbon removal in 2050 (disaggregated into bioenergy with carbon capture and storage, re- and afforestation, and other), and **c** the percentage change between 2020 and 2030 for all other indicators except for the electricity price, where the change between 2020 and 2025 is shown as electricity prices peak in 2025. The colored dots show the percentage change between 2020 and 2050. See Table 1 for the description of scenarios S1–S5.

In general, behavioral changes of consumers have entirely positive effects, corroborating earlier results[21,22]. They reduce not only the necessary $CO_2$ price, but also the residual emissions especially from non-$CO_2$ gases which are hard to reduce otherwise, and therefore the reliance on CDR. This leads to lower consumer prices, lower energy demand, and positive environmental and health effects. Though these changes result from individual consumer choices and are difficult to influence at sufficient breadth across society by policies, policies can support consumption choices by structural changes and political action to enable the uptake of low-carbon choices, e.g. by reducing barriers like lack of information[25].

None of the scenarios can achieve the climate target without CDR. To achieve GHG neutrality, residual emissions in 2050 in the range of 740-1180 $MtCO_2eq/yr$ must be balanced with CDR, including the 200 Mt $CO_2/yr$ residual emissions that remain uncompensated in the main scenarios. If CDR is limited due to limited CCS, residual emissions have to be reduced further, e.g. via lifestyle changes. If both, CDR and the reduction of residual emissions are limited, GHG neutrality cannot be achieved.

$CO_2$ prices above 100 €2020/t $CO_2$ in 2030 are needed in all scenarios to achieve the climate target. The required $CO_2$ price can be limited to this level by allowing technologies with limited social acceptance such as CCS, by regulatory measures such as banning combustion engines and oil and gas heating, or by changing consumer behavior towards a more sustainable lifestyle. If none of this occurs, very high $CO_2$ prices of over 450 €2020/t $CO_2$ will be needed as early as 2030. To achieve their climate target, the EU therefore needs to either

accept the use of technologies like CCS, facilitate comprehensive lifestyle changes, or accept very high costs. However, $CO_2$ prices occurring in the real world will differ from the prices derived in this study for two reasons. First, the model assumes a first-best solution without market failures, with full global cooperation, and with perfect foresight. Second, the $CO_2$ price depends on many assumptions, especially technology costs, which are still uncertain for a range of new technologies like many CDR options.

High $CO_2$ prices lead to substantial increases in energy prices and thus expenditures for many actors, but especially for poorer households. The impacts on poor households can be addressed, e.g. via redistribution of the $CO_2$ revenues, but this requires additional policies that are potentially difficult to implement. Nevertheless, a $CO_2$ price has the advantage that revenue is generated that can be redistributed to reduce or even convert regressive effects[18,19]. While removing limitations for technologies and changes in consumer choices reduce economic costs and therefore the $CO_2$ price, regulatory sector policies may reduce the $CO_2$ price, but still have economic costs if these sector policies do not address existing market failures. However, these economic costs do not show explicitly in the $CO_2$ price, and there is no additional fiscal income generated that could be used to mitigate distributional effects. It would be an important area for future research to quantify the economic costs associated with sector policies and compare them to the costs resulting from higher $CO_2$ prices. In the end, the choices between using technologies like CCS and accepting higher $CO_2$ prices, between changing consumer choices and accepting higher $CO_2$ prices as well as energy prices and greater reliance on CDR, and between very high $CO_2$ prices and more moderate $CO_2$ prices in combination with regulatory sector policies will be up to policymakers and societies.

## Methods

### REMIND-MAgPIE integrated assessment modelling framework

For this study, we use the global multi-regional energy-economy-land-climate model REMIND-MAgPIE 2.2-4.3.2[26–29] to derive cost-efficient emission and technology pathways. In addition to the REMIND model with version 2.2, we implemented several new policy options specifically tailored to Europe that represent the specific policies used in this study that are described in the introduction and in Supplementary Table 1. These changes, however, do not affect the fundamental model dynamics. REMIND-MAgPIE represents 12 subregions, namely the European Union including the United Kingdom, four individual countries (China, India, Japan, United States of America) and seven aggregate regions (Canada, Australia, New Zealand; Latin America; Middle-East and North Africa; non-EU Europe; other Asia; reforming economies; Sub-Saharan Africa).

### The REMIND model

REMIND[26] is an open source[27] global multi-regional general equilibrium economic growth model. The macroeconomic core of the model is hard-coupled to a detailed representation of the energy sector. The model assumes that economic agents (i.e., private and government investors in particular) have perfect foresight. For example, future price developments - especially the level of the $CO_2$ price - are anticipated. Under the premise that certain climate targets are achieved (here, residual emissions of 200 Mt $CO_2$eq/yr for Europe in 2050, as well as a global cumulative carbon budget of 500 Gt$CO_2$ from 2018 onwards, which is consistent with a 1.5 °C target), REMIND determines an intertemporal Pareto optimum of global welfare. This means that the different scenarios are not future projections, but possible economically optimal transformation paths based on a variety of assumptions.

To achieve the climate target, a wide range of different technologies is available to convert primary energy into useful energy in the end-use sectors. The energy sector is subdivided into the detailed end-use sectors of buildings, industry, and transport, as well as the electricity sector. Primary energy carriers (coal, gas, oil, biomass, uranium)

can be traded internationally. For the determination of bioenergy prices, as well as for the mapping of emissions from agriculture, REMIND is coupled to the land use model MAgPIE (see below).

In addition to technologies for energy production and conversion, there are technologies for $CO_2$ capture and utilization (e.g., for the production of synthetic fuels), as well as the option for $CO_2$ storage, needed e.g. for carbon dioxide removal technologies, specifically bioenergy with carbon capture and storage (BECCS) and direct air capture with carbon storage (DACCS).

The transformation measures listed above are implemented either explicitly as direct constraints (e.g. limits on CCS, bioenergy production or bioenergy trade; phase out of fossil heating), implicitly via costs (e.g. subsidies on electricity prices or hydrogen, or higher costs for curtailment and storage requirements for renewables as a proxy for inhibited wind energy expansion), or by changing exogenous assumptions (e.g. dietary changes, reduced energy demands, or modal switches in the transport sector).

System boundaries and limitations: Since REMIND is a global model, the spatial resolution is limited and the dynamics of individual countries from the EU + UK region are not represented. Model results such as demand, prices, and emissions are therefore mean or total values for the entire EU + UK region. Trade of secondary and useful energy carriers is not represented. For example, Europe cannot import hydrogen in the scenarios.

Furthermore, it is assumed that the policies implemented in the EU under the various scenarios do not directly affect the rest of the world. For example, other countries do not respond to electrification policies, which means that in certain scenarios, for example, the EU already has a very high share of electric cars, while countries such as the U.S. and China still largely rely on internal combustion vehicles. This in turn influences the European transformation via world market prices of primary energy sources.

### The MAgPIE model

MAgPIE (Model of Agricultural Production and its Impacts on the Environment) is a modular, open-source framework developed for modeling global land systems[28,29]. It integrates agro-economic and biophysical constraints within the land-use sector to project optimal spatial patterns of agricultural production under global scenarios until the end of the 21st century. Additionally, MAgPIE assesses the environmental implications of the land-use system. By doing so, it offers a comprehensive framework for investigating future pathways of land system transformation. The holistic approach considers the synergies and trade-offs between ecosystem services and sustainable development, allowing for a nuanced exploration of the complex interactions within the land-use system.

MAgPIE is a flexible global land system model that operates at multiple spatial resolutions. It encompasses three distinct spatial scales: First, World regions, which can be defined based on aggregated countries, allowing for a broader regional perspective; second, spatial clusters, identified by similar local characteristics, aggregating input data from a 0.5° x 0.5° spatial grid resolution; and third, grid-level with a resolution of 0.5°x0.5°, used for detailed biophysical inputs and disaggregation of outputs for land use patterns. As its input, MAgPIE incorporates detailed information on terrestrial carbon content, water availability, and potential crop and pasture yields from a global gridded vegetation and hydrology model LPJmL (Lund-Potsdam-Jena model with managed Land)[30]. The gridded inputs are integrated into a spatial cluster based on proximity in potential crop yield projections for the purpose of a non-linear program[31]. Furthermore, MAgPIE incorporates regional economic characteristics, such as agricultural product demand[32], technological advancements[33], and production costs.

MAgPIE is a partial equilibrium model of the agricultural sector with recursive dynamic optimization. It has a nonlinear objective function for minimization of global agricultural production costs to

the fulfillment of demand for agricultural products, subject to biophysical and socio-economic constraints MAgPIE determines agricultural demand for 25 food categories through initial demand trajectories obtained in a cross-country econometric regression analysis based on population and income development, demographic structures, and anthropometric characteristics. The demand reacts to income changes through iterative adjustments. The model encompasses various cost components, including factor requirements costs (capital, labor, fertilizer), irrigation costs (including investment in new infrastructure), land conversion costs, transportation costs to the nearest market, investment costs for yield-enhancing technological change (TC), and greenhouse gas emission tax under climate change mitigation scenarios.

Food demand is assessed by categorizing it into four distinct product groups: animal-source foods, empty calories (including sugar, oil, and alcohol), fruits and vegetables, and staples[32]. In scenarios where preference changes are considered, the estimation process focuses solely on caloric requirements, while exogenous assumptions are utilized to determine dietary composition and account for food waste. These assumptions are based on the transition towards a healthier and more sustainable diet (planetary health diet), as proposed by the EAT-Lancet Commission[20].

MAgPIE optimally allocates land use under competing demand for goods, forage, carbon storage, conservation, and environmental protection. Land use is broadly divided into cropland, forest, pasture, other natural land, and urban areas. The model calculates the following AFOLU greenhouse gas emissions: $CO_2$ resulting from land use change (includes change in soil and vegetation carbon stocks), $CH_4$ resulting from enteric fermentation, rangeland management, and rice cultivation, and $N_2O$ resulting from fertilization of agricultural soils. MAgPIE contains a fully dynamic and endogenous budget of the agricultural nitrogen cycle[34]. Nitrogen emissions from agricultural soils are calculated based on the IPCC 2006 Tier 1 method. This method does not distinguish between the different soil properties that emit $N_2O$, but applies a single emission factor for all soil types.

The model includes a selection of policies that can be used to achieve various sustainability goals. These include 1st and 2nd generation bioenergy, greenhouse gas emissions prices from land use change ($CO_2$) and agricultural land uses ($CH_4$, $N_2O$), land use regulations, REDD+ measures, reforestation, environmental management protection, and agricultural trade policies.

System Boundaries/Limitations: Economic assumptions are made at the level of 12 independent world regions, as in REMIND, with the EU27 + UK as a stand-alone region, which is the focus of this study. Therefore, direct interpretation of modeling results at the EU country level is not possible, and land use policy recommendations should be understood for the EU as a whole.

MAgPIE models the food systems and land use sector, including all relevant market drivers and biophysical characteristics of the land. However, some options and factors are not included in the model and therefore cannot be analyzed in this study. These include, for example, organic farming and various feedback effects of land management such as land degradation. The model includes the main options for mitigating GHG emissions in the land use sector (including afforestation, halting deforestation, bioenergy production, and less polluting agricultural production practices), but is not comprehensive in this regard. In particular, peatland and soil carbon management are not modeled in the scenarios presented. Therefore, there may be additional emission reduction potentials that go beyond this study.

### Coupling of REMIND and MAgPIE

The coupling of REMIND and MAgPIE was already described in Strefler et al. 2021[9]: "REMIND and MAgPIE are soft linked to derive scenarios with equilibrated bioenergy and emissions markets. In equilibrium, bioenergy demand patterns computed by REMIND are fulfilled in MAgPIE at the same bioenergy and emissions prices that the demand patterns were based on. Second, the emissions in REMIND emerging from pre-defined climate policy assumptions account for the GHG emissions from the land-use sector derived in MAgPIE under the emissions pricing and bioenergy use mandated by the same climate policy. The simultaneous equilibrium of bioenergy and emissions markets is established by an iteration of REMIND and MAgPIE simulations in which REMIND provides emissions prices and bioenergy demand to MAgPIE and receives land use emissions and bioenergy prices from MAgPIE in return.

The resource potential of bioenergy in REMIND is represented by regional bioenergy supply price curves that are updated (scaled) in the iteration process according to the price response of MAgPIE. The information is exchanged between REMIND and MAgPIE by region."

### Scenario design

The assumptions for these scenarios have been identified in a co-production process[11,13,14] with German stakeholders from policy, industry, and civil society. This process involves various iterations, captures the motivation and needs of a diverse set of stakeholders, and derives different scenarios that attempt to cater to these needs. In the stakeholder process, we collected a range of transformation measures for the five sectors power, transport, industry, buildings, agriculture, and land-use, many of which are still highly debated, e.g. concerning social acceptance and political feasibility, but also concerning justice and fairness or freedom, e.g. the balance between price signals and regulatory policies. There was a broad request for an explicit accounting for behavioral changes. Measures with a high level of agreement regarding their relevance and necessity were included with the same assumptions across all scenarios, e.g. expansion of renewable energy. Measures that are more debated were included with varying assumptions in the scenarios, e.g. the availability of carbon capture and storage (CCS), nuclear power, wind power, direct and indirect electrification of industrial processes, electro mobility, a phase-out of internal combustion engines and fossil-based heating, precision agriculture, and a demand shift towards lower energy demand in transport and buildings as well as a dietary shift towards less animal-based products[20].

The narratives were translated to scenarios that aimed to be both globally consistent with the 1.5 °C target and meet the EU climate target of net-zero GHG emissions in 2050 using REMIND-MAgPIE. All measures beyond carbon pricing were only applied to the EU in order to ensure comparability of scenarios, all other regions use carbon pricing only. Carbon prices are regionally differentiated to represent different abilities to pay. To achieve their climate target, the EU is neither allowed to buy international emission offsets, nor import bioenergy to avoid inducing land-use change emissions. The international dimension is important in order to consistently present scarcity or abundance of resources such as bioenergy or fossil fuels and the resulting prices, but also the consequences of technology diffusion.

## Data availability

The specific model runs and scenario data as well as plotting routines for this study are archived at Zenodo under a CC-BY-4.0 license upon publication and is available under https://doi.org/10.5281/zenodo.10552787. This includes the specific code used for this study as there have been some changes from REMIND v2.2.

## Code availability

The REMIND code is available under the GNU Affero General Public License, version 3 (AGPLv3) via GitHub (https://github.com/remindmodel/remind/releases/tag/v2.2.0). The code used in this study deviates slightly from version 2.2 and is therefore made available at Zenodo as described above. The technical model documentation is available under https://rse.pik-potsdam.de/doc/remind/2.1.3/. The source code and input data of MAgPIE v.4.3.2

(https://github.com/magpiemodel/magpie/releases/tag/v4.3.2) are openly available at https://zenodo.org/record/4624341. The technical model documentation is available at https://rse.pik-potsdam.de/doc/magpie/4.3.2/.

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

## Acknowledgements

The research leading to these results has received funding from the German Federal Ministry of Education and Research (BMBF, grant numbers 01LA1809A and 01LA1809B, DIPOL project, J.S., L.M., M.S., N.B., E.K., D.T.). This work was supported by the European Union's Horizon 2020 research and innovation program under grant number 821124561 (NAVIGATE project, F.H., E.K.) and by the BMBF under grant number 03SFK5A (ARIADNE project, R.P., M.P., R.R., M.R., G.L.).

## Author contributions

J.S. designed the research together with L.M., N.B., M.S., and E.K.; J.S., L.M., N.B., D.K., G.L., R.P., M.P., R.R., M.R. and E.K. contributed to developing the energy–economy modelling. M.S., F.H. and A.P. contributed to the land-use modelling. L.M. and M.S. performed scenario modelling. J.S., L.M. and M.S. performed data analysis and created the

figures. D.T. conceptualized the stakeholder dialogue and facilitated the co-creation process. J.S. wrote the paper with input and feedback from L.M., N.B., M.S., D.T., F.H., D.K., G.L., M.P., R.P., A.P., R.R., M.R., E.K.

## Funding

## Competing interests
The authors declare no competing interests.
