## [Peer Review File · Nature Communications]

REVIEWER COMMENTS

Reviewer #1 (Remarks to the Author):

Dear authors,

I congratulate you on this excellent manuscript, which I believe merits publication. It is high time that modelling take into account different policy approaches and unpacks their trade-offs (especially combinations of carbon pricing with other policy tools), and the manuscript does a superb job at precisely that. It is a welcome and necessary addition to the literature.

I would ask the authors to do just one thing: In an appendix, please detail the exact methods and results from the stakeholder communications in greater detail. I struggled to fully understand how this was done and how the results were incorporated in the model. In particular, I was surprised that the models did not include a "renewables only" scenario. In the current version, renewable energy and nuclear were either presented as restricted or not. What happens when one restricts nuclear energy, but not wind power?

Also, why did the author(s) omit a scenario that reaches carbon neutrality without carbon dioxide removal, but relaxing other constraints, such as some international transfers?

Reviewer #2 (Remarks to the Author):

Thank you for the opportunity to review this paper. It is a very ambitious work that addresses a relevant issue and advances knowledge in the field. The article provides an original contribution by incorporating three relevant dimensions for the achievement of the EU Net-zero target and leveraging stakeholders' opinions for building scenarios.

I see two aspects that can be improved in the paper:

The first relates to the construction of the scenarios. Extensive work was done to build those scenarios, but their particularities are not currently being exposed in the paper. More details on the scenarios' content is necessary to better understand the results. It seems that more specific criteria were adopted in each dimension of "technology & innovation", "political coordination", and

“behavioral change” than the dichotomies that are mentioned in the explanation of the scenarios. Table 1 is helpful, but it seems like those scenarios were more complex than just changes on those three criteria. The article would profit much from a section describing with more detail the elements that compose each one of those dimensions. For example: What were the different technologies considered, and how are they expected to behave in each scenario? The results mention different industries, but it is not clear how they were considered in each scenario.

The second relates to the discussion of the results and the conclusions the work provides. Those could be developed further, especially to include recommendations for policymakers. The authors hint at the conclusion of possible issues related to high CO₂ prices and poor households, but the CO₂ prices mentioned can also have a major impact on industries. In my view, such a high CO₂ price is not a desirable or feasible pathway, so more emphasis could be placed on what needs to be done to avoid it.

Reviewer #3 (Remarks to the Author):

The topic of this research is interesting, but major issues need to be addressed before publication.

1. The authors must use the impersonal style in writing the manuscript (without "we", "our study", etc.).
2. I recommend the authors to use a standard structure for the paper - introduction, literature review, method and data, discussions, conclusions
3. The authors must specify in the abstract the analysed period, method used and main conclusions of the study.
4. In the introduction, the authors must mention the objective of the study and the main contributions of the authors .
5. The literature review section needs to be inserted. In this section, the results of the most important studies in the field must be presented. The authors must demonstrate the existence of a research gap that justifies the completion of this study.
6. The discussion section needs to be inserted. The authors must interpret the results obtained in the context of similar studies that confirm or refute their results.
7. The authors must improve the conclusions section taking into account the results of the study.
8. The authors must improve the policy recommendations. Use specific statements rather than general phrases to indicate the policy options resulting from the main findings.
9. Limits of the research and future directions of research must be presented in the end of the paper

REFEREE REPORT(S):

Reviewer #1 (Remarks to the Author):

Dear authors,

I congratulate you on this excellent manuscript, which I believe merits publication. It is high time that modelling take into account different policy approaches and unpacks their trade-offs (especially combinations of carbon pricing with other policy tools), and the manuscript does a superb job at precisely that. It is a welcome and necessary addition to the literature.

We thank the reviewer for this positive assessment.

I would ask the authors to do just one thing: In an appendix, please detail the exact methods and results from the stakeholder communications in greater detail. I struggled to fully understand how this was done and how the results were incorporated in the model. In particular, I was surprised that the models did not include a "renewables only" scenario. In the current version, renewable energy and nuclear were either presented as restricted or not. What happens when one restricts nuclear energy, but not wind power?

Thank you pointing this out. We fully agree that more detail is useful, and therefore added figures and text on the impacts of the different measures in all dimensions in the Appendix. The new figure SI1b shows that nuclear power is hardly used, so that this specific limitation actually makes almost no difference. This is mainly due to the high costs of nuclear and the high potential of economically competitive alternatives.

In general, the reason to combine changes in several measures into one scenario was to keep the number of scenarios limited in order to not sacrifice readability. Including all relevant levers while keeping the number of scenarios as low as possible was one of the main challenges for the scenario design. We agree that scenarios where more single levers are varied would be interesting, but we tried to cluster them in a meaningful way. For this specific case, we limited all options where there is at least a perceived lack of public support in at least some countries, which includes wind power as well as nuclear energy.

We also added a section in the appendix on the stakeholder process.

Also, why did the author(s) omit a scenario that reaches carbon neutrality without carbon dioxide removal, but relaxing other constraints, such as some international transfers?

Thank you for raising this point. To our understanding, this is how the EU climate target is meant at the moment. Though this perception could be wrong, or at least it could change in the future, including international transfers would effectively mean watering down the EU climate target, and the level of transfers allowed for would be arbitrary. In this study, we were interested in the impacts of different strategies achieving the same level of ambition locally and therefore included only scenarios that achieve the emission neutrality target locally in the EU.

Reviewer #2 (Remarks to the Author):

Thank you for the opportunity to review this paper. It is a very ambitious work that addresses a relevant issue and advances knowledge in the field. The article provides an original contribution by incorporating three relevant dimensions for the achievement of the EU Net-zero target and leveraging stakeholders' opinions for building scenarios.

We thank the reviewer for this evaluation.

I see two aspects that can be improved in the paper:

The first relates to the construction of the scenarios. Extensive work was done to build those scenarios, but their particularities are not currently

being exposed in the paper. More details on the scenarios' content is necessary to better understand the results. It seems that more specific criteria were adopted in each dimension of "technology & innovation", "political coordination", and "behavioral change" than the dicotomies that are mentioned in the explanation of the scenarios. Table 1 is helpful, but it seems like those scenarios were more complex than just changes on those three criteria. The article would profit much from a session describing with more detail the elements that compose each one of those dimensions. For example: What were the different technologies considered, and how are they expected to behave in each scenario? The results mention different industries, but it is not clear how they were considered in each scenario. **The details of the scenarios are described in Table S1 in the Appendix. We now added a link to this table in the text. We also added more detail to the appendix, including figures that show the impacts of the specific measures and some more explanatory text.**

The second relates to the discussion of the results and the conclusions the work provides. Those could be developed further, especially to include recommendations for policymakers. The authors hint at the conclusion of possible issues related to high CO2 prices and poor households, but the CO2 prices mentioned can also have a major impact on industries. In my view, such a high CO2 price is not a desirable or feasible pathway, so more emphasis could be placed on what needs to be done to avoid it. **The reviewer raises an important point. However, the question whether high CO2 prices are feasible and desirable is a normative one. In this study, we do not attempt to answer this question, but rather outline different strategies that lead to different outcomes, which may be acceptable to different groups. The impacts of the different strategies should become clear, but in our view there is no clear recommendation, but the choice is up to policy and society. We added some text in the conclusions to make this point and the dilemma the EU is facing clearer. In particular we put more emphasis on the distinction between carbon prices and real economic costs.**

Reviewer #3 (Remarks to the Author):

The topic of this research is interesting, but major issues need to be addressed before publication.

1. The authors must use the impersonal style in writing the manuscript (without "we", "our study", etc.).

We politely disagree with the reviewer on this point. Personal language is widely used in many articles, including those in Nature Communications.

2. I recommend the authors to use a standard structure for the paper - introduction, literature review, method and data, discussions, conclusions **We included a heading "results" before the four subsections of the results part. Other than that, the structure as it is is a requirement by Nature Communications.**

3. The authors must specify in the abstract the analysed period, method used and main conclusions of the study.

In the abstract we mention that we analyse the time until 2050 using scenarios from a global multi-regional energy-economy-land-climate model that were co-designed with stakeholders. We also mention the main results, namely (1) without behavioral change but with restriction of technologies, the target of GHG neutrality in the EU could not be reached, (2) CO2 prices above 100 €/tCO2 in 2030 across all sectors are needed in all scenarios to achieve the emission target, and (3) the required CO2 price can increase to up to 450 €/tCO2 by 2030 if technologies are constrained, if no

complementary regulatory measures are implemented, and if changes in consumer behavior towards a more sustainable lifestyle do not materialize. Given the tight word limit of the abstract, we believe that these important points are all covered.

4. In the introduction, the authors must mention the objective of the study and the main contributions of the authors .

We mention the objective of the study at the end of the first paragraph of the introduction: "While the impacts of technology options on climate change mitigation strategies have been extensively studied, more comprehensive studies also including an interplay with different policies and behavioral changes across all sectors are much less common. In this study we aim to close this gap by bringing together different technology options, policy measures, and behavioral changes in coherent scenario set.". We also mention again at the end of the introduction that **"This paper provides new insights into the interplay between policies, technology availability, and behavioral change, and their impact on land and energy systems and on selected economic, social, and environmental indicators informing the feasibility of the net zero transition."**

Author contributions are mention in the specific section at the end of the manuscript, as it is required by Nature Communications. We believe this is good practice and it is widely adopted, so we would like to avoid duplicating this statement in the introduction.

5. The literature review section needs to inserted. In this section, the results of the most important studies in the field must be presented. The authors must demonstrate the existence of a research gap that justifies the completion of this study.

While we do not have a section that is specifically called "literature review", we present this important part in the first two paragraphs of the introduction. Also the research gap is demonstrated in the first paragraph of the introduction.

6. The discussion section needs to inserted. The authors must interpret the results obtained in the context of similar studies that confirm or refute their results.

The discussion including the interpretation of the results in the context of similar studies to the extent they are available is part of the section "discussion and conclusion".

7. The authors must improve the conclusions section taking in account the results of the study.

We summarize and interpret the results of the study in the beginning of the "discussion and conclusion" section.

8. The authors must improve the policy recommendations. Use specific statements rather than general phrases to indicate the policy options resulting from the main findings.

The reviewer raises an important point. In this study, we attempt to outline different strategies that lead to different outcomes, which may be acceptable to different groups. Therefore, there is no clear recommendation towards one of the scenarios, but the choice is up to policy and society. However, the impacts of the different strategies should become clear, which then allows for an informed choice. We added some text in the conclusions to make this point and the dilemma the EU is facing clearer.

9. Limits of the research and future directions of research must be presented in the end of the paper

We thank the reviewer for pointing this out. We have added limits of the study (especially regarding the uncertainties of the CO2 price) and future directions of research in the discussion section.

REVIEWERS' COMMENTS

Reviewer #1 (Remarks to the Author):

I thank the authors for their convincing revisions and additions and am now willing to recommend publication of the manuscript in the current form.

Reviewer #2 (Remarks to the Author):

The authors have thoroughly addressed my comments, and I have no further comments.

Wish them lots of success!

Reviewer #3 (Remarks to the Author):

The authors took into account the recommendations made by the reviewers and improved the manuscript. In this form, the manuscript can be published.